# Blank Spots in the Map of Human Skin: The Challenge for Xenotransplantation

**DOI:** 10.3390/ijms241612769

**Published:** 2023-08-14

**Authors:** Olga L. Cherkashina, Elena I. Morgun, Alexandra L. Rippa, Anastasiya V. Kosykh, Alexander V. Alekhnovich, Aleksey B. Stoliarzh, Vasiliy V. Terskikh, Ekaterina A. Vorotelyak, Ekaterina P. Kalabusheva

**Affiliations:** 1Laboratory of Cell Biology, Koltzov Institute of Developmental Biology, Russian Academy of Sciences, 119334 Moscow, Russia; 2Center for Precision Genome Editing and Genetic Technologies for Biomedicine, Pirogov Russian National Research Medical University, 117997 Moscow, Russia; 3Federal Government-Financed Institution “National Medical Research Center of High Medical Technologies n.a. A.A. Vishnevsky”, 143421 Krasnogorsk, Russia

**Keywords:** xenograft, xenotransplantation, humanized mice, human skin

## Abstract

Most of the knowledge about human skin homeostasis, development, wound healing, and diseases has been accumulated from human skin biopsy analysis by transferring from animal models and using different culture systems. Human-to-mouse xenografting is one of the fundamental approaches that allows the skin to be studied in vivo and evaluate the ongoing physiological processes in real time. Humanized animals permit the actual techniques for tracing cell fate, clonal analysis, genetic modifications, and drug discovery that could never be employed in humans. This review recapitulates the novel facts about mouse skin self-renewing, regeneration, and pathology, raises issues regarding the gaps in our understanding of the same options in human skin, and postulates the challenges for human skin xenografting.

## 1. Introduction

The skin is the largest organ in the human body; that is why it has always been the focus of attention in medicine and science, from antiquity to the present day. Human skin as well as laboratory animal skin investigations have revealed the layered skin structure, the events during the morphogenesis of skin and its derivates in embryogenesis, and the mechanisms involved in skin regeneration. Compared with animal models, the study of human skin has several restrictions. In basic research, human skin is taken as a biopsy from a single or several donors. This is suitable for detailed analysis of the skin structure in normal or pathological conditions, while at the same time resulting in the loss of the possibility to trace the changes during the time that is necessary for understanding the cell movements during self-renewal to maintain homeostasis or for analyzing the regeneration or pathogenesis progression. Another kind of problem is large genetic, age, and other individual differences between donors of biological material that could result in significant dispersion in the experimental data. There are several ways to overcome this barrier. One of them is the application of noninvasive methods to monitor the angiogenesis, wound closure, and fluctuations of several chemical indicators [1,2,3,4]. Alternatively, the single or serial biopsies could be excised after creating wounds in healthy donors under the control of the ethics committee [5,6,7,8]. This type of research is unique and cannot be used routinely. Numerous in vitro approaches and techniques have been developed: from single cell type monolayer cultures, complex multicellular 3D systems, including bioprinting and organs-on-chip, and explant cultures to study cell physiology, proliferation, differentiation, identification of cell-specific markers, and cell-to-cell interactions [9]. By favoring the wide spectra of in vitro models, the highly reproducible experiments could be performed, although it should be considered that the cell behavior would vary from the native skin due to loss of cell-type specific microenvironment. Developing single-cell and multi-omics technologies partially solves these issues by providing the transcriptional programs of every single cell in different cellular states at distinct points of the dynamic process to trace the cell fate [10,11,12,13]. Despite the latest insights into human skin structure and physiology, a plethora of knowledge has been accumulated in laboratory animal research.

Experiments that employ laboratory animals admit to applying an adequate sample size that is necessary for the statistical analysis and preventing the real effect from being missed. Animal experiments allow the use of one of the most powerful tools in investigative dermatology: genetic modifications and different chemicals to analyze proliferation, lipid uptake, lineage tracing, knockouts or knockdowns, etc. The mouse (Mus musculus) is the most widely used mammalian model organism; less research is performed on rats, guinea pigs, rabbits, pigs, primates, etc. Compared to that of mice, rat and rabbit skin is more similar to human skin in its biomechanical properties [14,15]. Porcine skin is the closest to humans in thickness, wound healing without contraction, and the composition of immune cells, although there are differences in the distribution of sweat glands and blood vessels. The rare use of these animals is associated with complications of housing conditions and the low availability of pig antibodies [16,17]. The guinea pig’s immune system more closely resembles that of a human which makes it a good object to study skin infections and autoimmune disorders compared to other animals [18,19]. However, not all traits of skin organization can be translated from animal experiments to humans: a detailed human skin investigation revealed differences in cell behavior and specific markers [20]. 

The xenotransplantation of human skin into laboratory animals is the way to overcome the limitations that have stood the test of time. Xenograft models have evolved in the field of studying normal as well as pathological human skin over the past 50 years [21]. Depending on the model application, research groups designed unique experimental procedures, including transplantation methods that allow for preserving the biggest graft area or full thickness, bearing the hair follicles (HFs), or skin pathology sites such as scars or wounds, combined with viral transfection, immune cell transplantation, microbiome maintenance, etc. The current review summarizes the questions that emerged from the knowledge about mouse skin and the possible ways to solve them based on developed and applied approaches regarding the existing models of human-to-mouse skin transplantation.

## 2. Basic Principles in Xenotransplantation 

One of the pioneering models of the transplantation of human skin into immunodeficient mice was described in 1973. The application of nude athymic mice allows maintaining the viability of grafts for 4–6 weeks, while the skin grafting to wild-type mice resulted in rapid graft rejection. The experimental design was not optimal: some animals developed inflammation after transplantation despite the immunosuppression [22]. However, this and similar experiments provided promising results to expand the knowledge of human skin physiology. Thus, since the 1970s, human skin transplantation to immunodeficient mice has become a popular model. Briefly, the main principles for selecting a xenotransplantation model are designated in Figure 1.

### 2.1. Animals

The development of the appropriate mouse strain requires maximal immunity suppression and minimal risks of graft rejection. BALB/c or C/B-17 mice are usually used as a background. The first mouse strain that gave rise to the xenotransplantation method was BALB/c-nu/nu (Nude), which lacks T-cell immunity but retains B-cells, natural killer (NK) cells, and macrophages, therefore frequently resulting in graft rejection or scarring [23,24,25]. 

To efficiently suppress mouse immunity, T- and B-cells should be removed. Now, in most experiments mice with autosomal recessive mutation in Prkdc^scid^ alleles (SCID mutation) are used for human skin transplantation. SCID mice have impaired T- and B- lymphocyte development [26,27]. Rag1 or Rag2 knockout (Rag^−/−^) which leads to T- and B- lymphocyte arrest in bone marrow may also be used [28].

Another important component of immunity are NK-cells. NOD strain is characterized by deficient NK-cell function and may be used as a background for SCID or Rag^−/−^ mutations. NK-cell deficiency may also be caused by bg mutation [29,30]. Mutation in the interleukin-2 gamma chain receptor (IL2rγ) leads to the disruption of signaling pathways that are involved in hematopoietic cell development and NK-cell differentiation. Its application significantly reduces the graft infiltration and may be combined with SCID [31,32,33] or with Rag^−/−^ mutations [34,35].

Different combinations of mentioned background strains and mutations may be used to improve mouse immune tolerance. For example, the NIH-III mouse strain has a nude background and also possesses a xid mutation that affects the maturation of T-independent B-lymphocytes and a bg mutation [29,30]. One of the popular strains is NOD-scid IL2rγ^null^ or NSG that has a NOD background, SCID mutation that affects T- and B-cells, and mutation in IL2rγ that affects other immunity components [31,32,33].

Several research groups prefer to use rat strains versus mice [36,37]. Rat skin morphology is closer to human skin as compared to mouse models, and the rat lifespan is longer than that of mice, thus providing more prolonged experiments. 

The experimental design requires an individual approach to the selection of an appropriate mouse strain. The comparison between SCID/bg and NSG has shown that human skin morphology in SCID/bg mice is better reproduced compared with NSG. At the same time, co-transplantation with hematopoietic cells is more efficient in the NSG strain [38]. Tumor xenograft growth dynamics are also different between various mouse strains. In such cases, a panel of several mouse strains may be used [39].

The reconstruction of some human skin processes is equally successful by applying different mouse strains. For example, the pathogenesis of varicella-zoster virus (VZV) did not differ between xenografts in SCID and nude mice [23] or SCID and NOD-SCID [27]. 

There are a small number of studies comparing skin xenografting between several mouse strains perhaps due to the technical difficulties of model construction and high costs. Therefore, the choice of specific mouse strain may be based on previous research where similar conditions were studied.

### 2.2. Where to Place the Graft?

There are two main ways of graft positioning regarding mouse skin: the first is placing the human skin into the full-thickness incision on the mouse back (cutaneous transplantation), and the second is transplantation under the skin (subcutaneous transplantation). Lorenz identified that subcutaneous human fetal skin grafting preserved its intrinsic features of scarless regeneration while cutaneous transplantation resulted in the acquisition of adult skin regeneration [40]. A recent study demonstrated the transformation of fetal skin into adult-like skin 10 weeks after cutaneous grafting in NSG mice [36]. By the way, several studies use the intermediate option when human skin grafts are covered by live [41] or decellularized [42] mouse skin after transplantation, which eventually leads to the complete incorporation of the graft into mouse skin.

### 2.3. The Size of Grafts

Planning the experiments, the balance between the size and functionality of the grafts should be taken into account. First of all, the skin structures under consideration should be determined, and the technical features elaborated to maintain their viability. Split-thickness skin grafts do not bear lower dermis, subcutaneous fat, or HFs, which prevent necrosis due to the accessibility of newly grown vessels and allow for a maximal area. These graft types are suitable for epidermis investigation, wound and scars modeling, studying skin immunology, intradermal cell transplantation, microbiome, graft rejection, etc. [36,42,43,44,45,46]. Full-thickness xenografts are required for dermis, subcutaneous fat, and angiogenesis, as well as HF and sebaceous gland investigations. HF grafting is often performed as an HF unit or punch biopsies from 3 to 6 mm^2^ area [47,48,49]. These basic principles of xenograft size selection are not designed for the research of an interfollicular area of the skin. To address this, fetal skin could be applied because it maintains hair growth and allows for a larger graft area [36,50]. Adult full-thickness skin grafts must be of a certain shape to prevent necrosis and be accessible for angiogenesis [41].

### 2.4. Skin Source 

The benefit of fetal skin besides its high regeneration is immune privilege which allows for the larger size of graft compared to the adult skin [36,51]. As mentioned, depending on the grafting method it could reproduce the features of fetal or adult skin [40]. Summarizing, fetal xenografts are the best object for human skin investigation, being the least accessible source of biological material. Foreskin grafting is the most common in human-to-mouse skin transplantation experiment, combining high regeneration and good availability [29,31,42,52,53]. Adult skin is appropriate to study normal skin physiology, regeneration, and disease modelling [47,54,55]. For several skin pathologies, a skin lesion biopsy can be used for transplantation [56,57].

### 2.5. Co-Transplantation

Co-transplantation of human skin grafts with different cell types, tissues, and organs could be practiced for completing the native environment for human skin or for analyzing the influence of supplemented materials on skin maintenance. Reconstruction of the human cutaneous immune system is one of the challenges of the humanized mouse method. Starting with the co-transplantation of cytotoxic cells to achieve autoimmune conditions [58,59,60], this approach has now developed into joint grafting of human skin with autologous lymphoid tissues and hematopoietic stem cells [36]. 

The addition of several cell types promotes better survival of the human grafts. The preliminary transplantation of hair follicle-specific dermal cells allows for the reconstruction of dissected hair follicles that are grafted afterward [30]. Injecting adipose-derived stem cells into full-thickness grafts decreased the inflammatory response, promoted angiogenesis, and prevented scarring after transplantation [37,61]. Dermal progenitor cell transplantation beneath a split-thickness graft restores the dermis structures and improves the mechanical properties of the skin [43]. 

Cells can be used as carriers of viral particles. For example, to infect a graft with the VZV, a transfected suspension of human lung or skin fibroblast cells is administered intradermally by injecting the suspension into the dermis or applying it to an incised skin area [62,63].

### 2.6. Disease Modeling

The pathological processes of reproduction and relevant drug testing are the most common application of xenotransplantation [21]. Grafts are used for acute wounding [64,65,66,67]. Compression devices are used to replicate pressure ulcers [68,69]. Several xenograft models are designed to study hypertrophic scars and verify the effects of potential cell [70], genetic [57], and drug therapy. By the way, scarring and ischemia often appear spontaneously after skin grafting, and this is often used as a wound [71] and scarring model [70].

Transplantation of diseased skin is a way to study hypertrophic scars [56,57], but it is not suitable for disorders of an autoimmune nature. The biopsies that were taken from patients with alopecia areata and alopecia universalis did not reproduce the pathogenesis after transplantation [72,73]. T-lymphocytes that were cultured with HF homogenates restored the phenotype of alopecia areata pathology [74]. Psoriatic skin grafting leads to regression of the epidermal thickness of the transplants with the loss of HLA-DR expression and a slight decrease in ICAM-1 expression [60,75]. Activated autologous T-cells induce the psoriatic conversion of the graft [76]. To address this complication, skin lesions could be induced in healthy skin. IL-2-activated peripheral blood mononuclear cells injection causes symptoms of alopecia areata [47,58,77]. 

Human skin xenotransplantation has been used to study the pathogenesis of different viral strains, to analyze the effect of genetic alterations on infectivity [26,78], and to test antiviral drugs [52]. The use of animals is often not possible due to resistance to human viruses or, on the contrary, due to high lethality [79]. Xenografts could be obtained from infected patients [80] and treated with viral particles before [81] or after xenotransplantation [23,27,52,82]. In some cases, animals that have been transplanted with human skin are directly infected [29,62]. Studies of xenografts have shown that viruses have different tropisms for certain tissues. VZV penetrates deep into the dermis and often provokes scarring; human cytomegalovirus also infects the dermis more often; and herpes simplex virus 1 (HSV) infection is limited to the epidermis [83,84]. VZV, unlike HSV, can also infect lymphocytes, and that is why the rash spreads throughout the body [83]. 

### 2.7. Xenotransplantation Restrictions

Human skin predominantly returns to its homeostatic conditions after xenografting; however, several major changes in skin physiology and histology occur. Despite the formation of a clear boundary between mouse and human skin [27,41], partial tissue replacement takes place. Mouse keratinocytes could be found in grafted epidermis [85], endothelial cells were partially replaced with mouse ones [31,86], and HF dermal papilla may also contain mouse cells [30]. Mouse cells participate in human skin regeneration in an acute wound model [40]. Resident CD4^+^CD103^+^ human T-cells can leave the transplant and migrate to the mouse spleen [87].

After transplantation, human skin acquires a fibrotic phenotype [46,88], epidermal thickening and hyperkeratosis which do not recover after 3–6 months [41,45]. HFs enter dystrophic catagen, the hair cycle stage of regression [49]. Moreover, HFs could lose pigmentation and demonstrate certain abnormalities in their morphology. 

## 3. Actual Questions from Mouse to Human Skin

Mus musculus is a frequently used mammalian model organism for investigative dermatology. First, mouse skin research does not have the same ethical restrictions as human research, thereby allowing a wider spectrum of experimental procedures. Second, the mouse is the most popular mammal for genetic modifications. The skin of mice and humans has a common structural plan, so the processes occurring in mice can indeed be partially extrapolated to humans. Common features include a three-layered structure: the epidermis, underlying dermis, and subcutaneous fat; the presence of HFs and sebaceous glands. The differences between mouse and human skin are briefly formulated in Table 1.

### 3.1. Epidermis

The epidermis comprises five layers: stratum basale (the deepest portion of the epidermis), stratum spinosum, stratum granulosum, stratum lucidum, and stratum corneum (the most superficial portion of the epidermis). The human epidermis is composed of 5 to 10 cell layers, whereas murine skin contains only 2 or 3 and possesses more rapid self-renewal. The proliferation of basal keratinocytes provides sufficient cell mass that moves outward and gives rise to all the epidermal layers. The first question addressed was whether all proliferating cells are equivalent. The exploration of label-retaining cells (LRC) and mathematical modeling founded the stem/transit-amplifying (TA) cell theory: the stem cell rarely divides asymmetrically into one stem cell and one TA cell that multiplies, providing further differentiated cells. Studying the mouse epidermis, Potten [89] has developed the epidermal proliferation unit (EPU) concept. The interfollicular epidermis structure appears as a “parquet” of hexagons of EPUs, residing in the base of EPU label-retaining slow cycling stem cells providing TA-cells, the descendants of which produce a column of differentiated cells. Lately, the analysis of the epidermis structure of the genetically modified Confetti mice revealed that individual clone’s shapes are arbitrary and do not match regular hexagons due to progenitor cells spreading along the basal lamina [90]. The observation of human epidermal stem cells was complicated because of the impossibility of applying DNA labels or genetic modifications. The primary knowledge was obtained from keratinocyte culture: epidermal cells produced the different clones in vitro; stem cells formed large, self-renewing holoclones; whereas TA cells produced abortive clones because they divided a small number of times and underwent terminal differentiation [91,92]. Combining the xenografting approach with viral transfection visualized the epidermal clones directly in the human epidermis [93]. While the mouse dermal–epidermis junction is flat, the human one is waved, with dermal papillae that are separated by rete ridges. It was assumed that stem cell location could depend on the position of rete ridges; however, Ghazizadeh and Taichman did not reveal the specific position of the epidermal clone’s origination.

For a long time, mathematical modeling predicted the existence of a single type of interfollicular epidermal stem cell, and the discussions were related to the proportions of the types of its divisions [94,95,96,97,98]. In 2016, two research groups published evidence of the presence of two populations of interfollicular epidermal stem cells: slow and rapidly cycling. Studying the mouse tail epidermis, Tumbar’s group identified that rapidly cycling non-LRCs are located in the tail scale area, while slow-cycling LRCs reside in the interscale [99]. Lineage-tracing, wound healing, and mathematical modeling approaches confirmed that these populations possess exact stem cell features and are not stem and progenitor cell populations. The determination of LRC- and non-LRC-specific markers allows identification of the same areas not only in the tail but also in the back mouse skin. Dlx1-positive LRCs have a tendency to be close to the hair follicle, while Slc1a3-positive non-LRCs are located in the interfollicular area. Another study also identified different types of epidermal clones by calculating their size based on the epidermal multicolor label of the mouse back epidermis [100]. Similar to Tumbar’s investigation, they explored the dependence of different stem cell positions on the HFs. Despite the previous group, Roy postulated that slow-cycling cells are distant from hair follicles while rapidly dividing cells are attached to them. Roy also identified that the frequency of proliferation of HF-attached clones is related to the stage of the hair cycle.

For humans, the basic principles of stem cell division were calculated using single cell transcriptomics data with further bioinformatical processing [13,101,102,103]. Tumbar’s group found the same stem cell segregation in human epidermis using verified cell markers. Sox6/Vamp1/Col17a interscale-like cells were found on the top of dermal papillae, while Slc1a3/K15 scale-like residues were found at the base of rete ridges [104]. The application of two databases previously published for single cell RNA-seq from human epidermis [105,106] with the following transcriptomic lineage trajectory maps suggested the existence of two distinct paths of interfollicular epidermis basal cell differentiation. Unexpectedly, there was no data confirming the differences in proliferation velocity in putative LRC and non-LRC areas of human skin. In a latter publication for identifying the interscale-like cells in human skin collagen, XVII was used as one of the markers. Col17a1 contributes to defining mouse tail scale shapes and human skin microtopography [107]. Mouse and human epidermal keratinocytes, which are positive for col17, possess a high clonogenic potential [108]. Driving cell competition, epidermal clones that express high levels of Col17a1, which divide symmetrically, outcompete and eliminate adjacent stressed clones that express low levels of Col17a1, which divide asymmetrically [108]. These indirectly indicate the correspondence of human interscale-like cells to LRCs.

The lack of spatial information of the different cell type localization and their proliferative potential could be compensated by the xenografting approach.

### 3.2. Dermis

The dermis is composed of two layers based on location, cellular density, and extracellular matrix composition. Dermal fibroblasts have been usually discussed as belonging to the upper papillary layer or lower reticular layer. Genetically designed mouse strains were used for lineage-tracing analysis that led to a careful description of mouse dermal fibroblast development and its involvement in wound healing. [20,109,110]. Briefly, a common fibroblast progenitor, positive for Dlk1 and Lrig, gives rise to Blimp1^+^Dlk1^−^lrig1^+^ papillary fibroblast progenitor and Blimp1^−^Dlk1^+^ reticular fibroblast progenitor. The former differentiates into papillary fibroblasts with a CD26^+^Blimp1^−^lrig1^+^ phenotype, while arrector pili muscle cells possess an Itga8^+^CD26^−^ phenotype. The latter produces reticular dermis with Dlk1^+^Sca1^−^ fibroblasts and Dlk1^+/−^Sca1^+^ hypodermal fibroblasts that further differentiated into adipocytes. HF dermal papilla cells develop from a common fibroblasts progenitor for primary hair follicles (Sox2^+^), and from papillary fibroblast progenitors for other follicle types (Sox2^−^). Unfortunately, the investigation of human fibroblasts was complicated by the absence of verified markers of papillary and reticular cells.

Significant progress has been achieved with elaboration of a single-cell approach. Two articles related to human dermal fibroblast diversity were published in 2018. The Watt group [111] started the identification of putative papillary and reticular fibroblast markers by bulk sequencing. The most enriched markers of papillary fibroblasts were Col6a5, Col23a1, and several members of the WNT pathway. Lower dermis upregulates secretoglobin superfamily genes and CD39 compared to the papillary dermis. All the markers were verified using immunostaining; positive cell groups possess different localizations and different effects on epidermal keratinocytes. Following single cell analysis, four subpopulations of dermal fibroblasts were revealed and it was confirmed that these different populations clustered in separate groups, although it was pointed out that fibroblast identity is not restricted by spatial compartmentalization within the dermis. The Tabib research [112] started straight from single cell RNA-seq and clustered the dermal fibroblasts into two large (Sfrp2/Dpp4 and Fmo1/Lsp1) and five minor groups. Immunohistochemical staining demonstrated that major populations did not correspond to papillary and reticular dermis; cells were widespread through whole dermis. The similar inconsistence of the fibroblast classification to dermal layers was stated in a Vorstandlechner study [113]. The next investigation [114] divided dermal fibroblasts into four groups: secretory–papillary, secretory–reticular, mesenchymal, and pro-inflammatory fibroblasts. Mapping the groups on the skin sections was conducted using RNA-Fish. Secretory–papillary and reticular cells were matched in the same layers; mesenchymal fibroblasts were identified in the hair follicle; and pro-inflammatory fibroblasts surrounded the blood vessels. However, these identified groups did not correspond to previously published data by markers expression. At this time, the reproducibility of the results became under consideration [115,116]. Ascensión reanalyzed previously published datasets and established the existence of common groups between different studies. Three major clusters were checked by immunostaining to define the localization. The first group is distributed throughout the whole dermis; the second and third ones are located in the HF dermal papilla and perivascular area, respectively. Clusters that corresponded to papillary and reticular dermis were not distinguished.

A similar situation appeared in the analysis of single-cell lipidomes: while papillary and reticular dermal layers are well-distinguished in skin slices, these fibroblast populations overlap in bioinformatic plots [117].

Thereby, before the single cell era, the key question was the discovery of papillary and reticular fibroblast-specific markers; now it concerns the existence of any differences between these subpopulations. It highlights the significance of cell fate tracing experiments to complete our understanding of the human skin dermis structure.

### 3.3. Hair Follicle

A HF is a skin mini-organ that produces hair shafts. A mouse has three waves of HF morphogenesis that result in the presence of four types of HF types: guard, awl, auchene, and zigzag hair, and also the special type—vibrissa follicles [118]. A Human has two generations of HFs: the embryo-type lanugo hairs and the adult, which are divided into terminal hairs in the scalp and beard, pubic/axillary hairs, and vellus hairs throughout the body [119]. The HF is composed of the epithelial part which includes the hair shaft, inner, and outer root sheaths. The inner root sheath grows along with the hair shaft, while the outer root sheath is a continuation of the interfollicular epidermis. The outer root sheath contains the bulge area, where the HF epithelial stem cells are located. Their descendants move downward and repopulate the hair matrix in the hair bulb, which gives rise to the hair shaft and inner root sheath. The HF epithelial compartment is one of the most multifaceted and specialized populations between the epithelia [120]. The mesenchymal part of an HF represents the dermal papilla that is embedded in a hair bulb and connective tissue sheath surrounding the hair follicle [121].

HFs are periodically regenerated during the hair cycle, which consists of anagen (growth phase), catagen (regression phase), and telogen (resting phase). While the mouse hair cycle duration is several weeks, for human hair follicles it takes several years. Additionally, the cycling of mouse HFs is synchronous, especially in the first few months after birth; however, human HFs cycle independently. Mostly, hair cycling is controlled by a crosstalk between dermal papilla and bulge. These structures produce the batch of growth factors and cytokines that regulate the HF epidermal stem cell quiescence, activation, proliferation, and further differentiation. These signaling molecules include WNT, BMP, SHH, FGF, and other growth factor families, which have been studied for a long time and are carefully systematized in several reviews [118,122,123]. Bulge stem cells are multipotent versus unipotent interfollicular epidermal stem cells. Besides the regular reconstruction of HF structures that disappeared through the catagen, they are able to encourage interfollicular epithelization after wound healing. This fact is determined for mouse skin; for human skin, this is still addressed.

In the last decade, the new mouse hair cycle controlling cells have been discussed. Dermal white adipose tissue (DWAT) cycles together with hair follicles: anagen onset induces adipocyte proliferation with subsequent lipogenesis, while catagen initiation causes lipolysis [124,125]. DWAT secretes BMP2 [126], PDGF [127], Follistatin, DKK1, and SFRP4 [128] which orchestrate the hair cycling. Aged DWAT cells produce FGF5 and CXCL1, which result in hair follicle malfunction [129]. Thus, these cells could be important in the same way for human hair follicle cycling or impaired hair growth.

Using human scalp skin biopsies and HF organ culture, it was found that DWAT cells undergo similar hair cycle-dependent changes in vivo and produce HGF that is necessary for stimulating scalp HF growth and pigmentation [130] There are plenty of humanized mouse models to study HF regeneration [47,48,49]. After the grafting, HFs enter dystrophic catagen and then initiate the growth phase, anagen. This method allows one to observe the fluctuations of adipocytes size during hair follicle regeneration [131]. This is probably the best way to understand the involvement of DWAT in human hair cycle regulation and disease.

The HF possesses its own distinctive immune system, violations of which lead to different types of alopecia. HF cells crosstalk with immune cells ensuring proper hair growth and protection against autoimmunity. Many facts have been accumulated about the involvement of immune cells, including T-cells, macrophages, and mast cells, in the regulation of the cycle [132,133,134]. Mast cells reside close to the hair follicles and possibly regulate hair follicle epithelial cell proliferation via cytokine secretion [135,136,137]. Regulatory T-cells augment HF stem cell proliferation and differentiation [138]. Activation of γδT-cells results in the proliferation of hair follicle stem cells [139]. Not all the cell types persist in human skin or keep the same functions. However, it is well known that immune system abnormalities lead to a set of disorders, so need a detailed investigation.

### 3.4. Wound Healing

Wound healing processes are very different between mice and humans [140,141]. Mouse skin wound closure occurs to a greater extent due to contraction of the skin and fusion of the edges of the damage, while human contraction is almost not pronounced due to the tight contact of the dermis with the underlying tissues. Another mouse wound healing feature is the appearance of the newly formed HFs (Wound-Induced HF Neogenesis, WIHN) instead of scarring in certain conditions [142,143]. Therefore, to approximate human wound healing, researchers resort to various methods, such as splinting to exclude contraction [144], creating an ischemic wound [145,146], or the creation of non-healing wounds in diabetic mice [147].

Dermal fibroblasts are one of the most important skin cell populations, promoting wound closure or extensive fibrotic changes under pathology. To study the dynamics of cell migration into wound beds, linage-tracing experiments were performed. It had been shown that the initial wave of dermal repair following wounding is mediated by lower lineage fibroblasts; by contrast, upper dermal fibroblasts are recruited during subsequent wound re-epithelialization [109]. One of the dermal fibroblast populations, the Engrailed 1 (En1)/CD26-positive cells, is enriched during wound healing and is involved in fibrosis progression [148]. En1/CD26-positive cells have dual origins: the first population originated even in embryogenesis, while the second arises from En1-negative cells via YAP1 signaling in adult skin in cases of wounding [149,150]. The elimination of these fibroblasts by employing specific small-molecule inhibitors or genetic knockouts resulted in the absence of scarring and substantial hair growth by 30 days [149,150]. The deficiency of CD26 results in delayed and reduced graft immune rejection after murine allogeneic skin transplantation, which suggests the involvement of En1/CD26-cells in inflammation and immune responses [151]. However, several single-cell transcriptomics studies, including spatial transcriptomics, have not pointed to En1/CD26-cells as major drivers of wound healing [152,153].

The number of CD26-positive cells increases in human skin after wounding in vivo [8]. In human neonatal foreskin xenografts, CD26-positive cells persist during wound healing [31]. A human skin reconstruction experiment in a mouse model demonstrated the importance of CD26-positive versus CD26-knockout fibroblasts for proper epidermal development, dermal reconstruction, and capillary growth [154]. The investigations of different skin pathologies do not highlight a CD26-positive cell population [155,156,157]. Thereby, this population could become a good target for anti-fibrotic drug development; nevertheless, first of all, the functions of these cells in human skin should be determined.

DWAT is supposed to be one of the major players during skin regeneration. Abrogation of injury-induced dermal adipocyte lipolysis results in reduced numbers of inflammatory macrophages, delayed repair [158], and an extensive extracellular matrix deposition during fibrosis [159]. It is interesting that CD26 could be one of the regulators of lipid metabolism during skin regeneration [159]. DWAT cells are directly involved in wound regeneration by replenishing myofibroblast pulls after wounding [158]; at the same time, myofibroblasts differentiate into skin adipocytes if the WIHN initiates after wound closure [160].

WIHN is one of the most important events in scarless mouse skin regeneration that is absent in human skin. Its initiation depends on the activation of several signaling pathways including YAP1 [149], WNT, and FGF9. The latter growth factor is produced by γδT-cells [134,161]. The low propensity for WIHN in human skin could be related to γδT cells deficiency. Human skin xenografting allows for the reproduction of all the conditions to verify this assumption.

## 4. New Methods in Xenografting to Close the Gaps in Skin Structure Knowledge

The analogy between the mouse and human skin structure indicates several significant knowledge gaps regarding our understanding of human skin cell composition and behavior.

Despite some discrepancies, the mouse and human epidermis contain a similar set of cell populations including LRCs and non-LRCs. Their distribution in the mouse epidermis depends on the distance from HFs, while in human skin it depends on rete ridges and inter-ridge domains [100,104]. Whether human HFs influence the spatial organization of LRCs in the epidermis or possess some other impact on epidermis maintenance and regeneration, as in mouse skin, remains questionable. Lineage-tracing experiments revealed that mouse HF bulge epithelial stem cells are involved in skin epithelialization after injury [162]; human HFs could contribute to epidermal regeneration similarly. Identification of epidermal stem cell populations and the detailed characterization of their molecular signatures and the localization throughout the epidermis is extremely important for our understanding of the wound healing processes and developing the cell therapy to increase the regeneration [163,164].

The gap in our knowledge between the human and mouse dermis is much more impressive. The separation of dermal fibroblast subpopulations is possible only using surgical instruments that assume the inability to obtain the pure population. Single-cell RNA-seq, one of the most powerful approaches in human skin research, did not establish the differences in expression profiles between two major dermal fibroblast populations: papillary and reticular [115]. Thus, the investigation of age-related changes [165], disease-associated abnormalities [166,167], and the impact of fibroblast heterogeneity in the epidermal stem cell niche is sophisticated.

DWAT is known to participate in HF cycle regulation as well as in skin regeneration after injury [123]. Adipose tissue mesenchymal stem cells-derived exosomes are known to stimulate fibroblast and keratinocyte proliferation, induce collagen remodeling, promote angiogenesis, and reduce inflammation [168]. However, in vivo studies of human material are needed to confirm their safety and efficiency as therapeutic agents.

A mouse skin investigation revealed the effects of resident immune cells on skin physiological functions. To reproduce human skin immunity in xenografts, a mouse immune system is suppressed and replaced by human cells. Studying the autoimmune disorders and immune barrier formation in a co-transplantation model, human T-cells [169,170,171], NK-cells [172], Peripheral blood mononuclear cells [58], as well as immune tissues such as a spleen, a thymus, and a liver have been transplanted along with skin. All the known or newly identified types of skin immune cells could be added to the xenograft. This is also the way to simulate the mouse immune system in human skin. It is interesting in the context of recently identified unique functions of mouse immune cells, in particular the involvement of γδT, aβT, and DETCs in hair cycle regulation, WIHN, and scarless wound healing [161,173].

Xenografts could contain all the cell types of human skin. Modern genetically modified immunodeficient mouse strains allow for the maintenance of up to 6 cm^2^ of human skin transplants for a year without rejection [21]. For the cell types that are depleted as a result of withdrawing from the human organism, such as immune and endothelial cells, there is a possibility to isolate and inject the cell suspension directly into the graft or into the recipient animal [174], or to transplant the tissue that produces the necessary cells [36]. However, the small number of cell types of interest should be considered when planning further experimental procedures due to the inability to employ several approaches [174].

Xenografting allows the application of vital tracers and chemicals. Nucleotide analogs (BrdU, EdU, IdU) can be used for label-retaining or label-dilution experiments [175]; fatty acid analogues, for example, BODIPY, can be used to analyze the DWAT metabolism [124]. Lentiviral particles bearing fluorescent proteins proved their efficiency in xenograft epidermis transfection and could be helpful for clonal analysis and cell tracing experiments [93,176].

Skin transfection could also be suitable for genetic alterations in gene expression, which are obligatory for deciphering the signaling cascades involved in homeostasis maintenance and regeneration after injury, defining intracellular protein functions, and exploring the potential targets for the genetic therapy of skin disorders. For example, siRNA has already been used in human skin xenografts [57,176]. While lentiviral transfection is suitable for highly proliferative epidermal cells, adeno- and adeno-associated viruses could be employed for quiescent cells [177]. Hypothetically, the further generation of viral transfection could be based on tissue-specific viruses, such as VZV or cytomegalovirus for dermal targeting, as was realized for orthoreovirus implementation for cancer xenografts [178]. There are several alternative techniques to deliver the genetic material, such as packaged with Histidine Lysine Polymer [57], electrotransfer [179], and microneedling with vectorless (naked) plasmid DNA [180].

Skin xenotransplantation allows for studying changes at different terms after grafting. Combining it with cell tracing methods will give us the possibility to describe the dynamical changes as it has already been conducted in mice.

## 5. Conclusions

In this review, we highlighted several issues related to human skin-specific markers, proliferation potential, origin, functions, and interactions with other cell types and skin structures. The lack of answers to these questions arises due to the limited spectrum of available methods for human skin research versus mouse skin. Human skin xenografting compensates for these differences. Combining with the single cell approach, it could significantly promote our knowledge about human skin development, self-renewal, and regeneration.

## Figures and Tables

**Figure 1 ijms-24-12769-f001:**
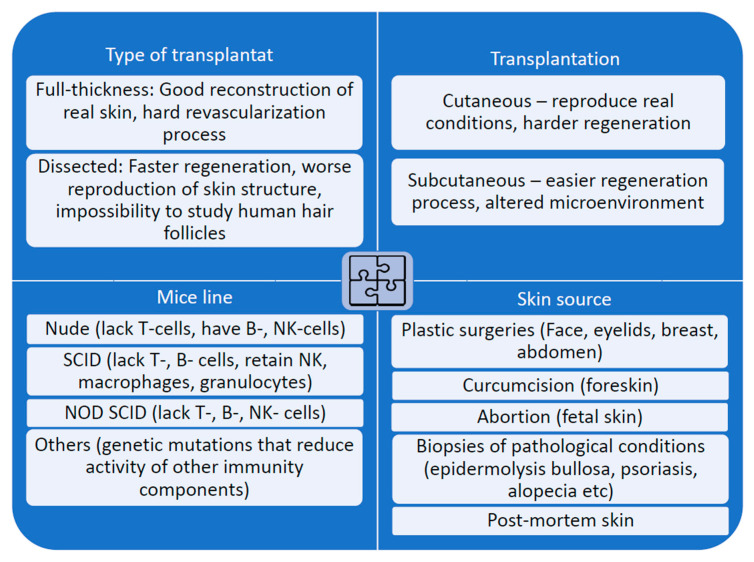
Different aspects of xenograft construction: it is important to choose an optimal combination of components depending on the experiment conditions and desired parameters of the xenograft.

**Table 1 ijms-24-12769-t001:** Comparison between mouse and human skin. LRC—label retaining cells, DETC—dendritic epidermal T-cells.

	Mouse	Human
Epidermis	2–3 cell layers LRC and non-LRC areas related to scale and interscale in a tail and HFs in other skin	5–10 cell layers LRC and non-LRC areas related to rete ridges
Dermis	Thickness 0.25–0.7 mm	Thickness 0.5–6 mm
Adipose tissue	Between the dermis and panniculus carnosus	Around pilocebaceous units
Panniculus carnosus	Present, promotes contractions and rapid healing	Only present as platysma (neck), palmaris brevis (hand), dartos muscle (scrotum), has no functional significance
Hair follicles	Four types of HFs: guard, awl, auchene, and zigzag Anagen duration 1–3 weeks HF noegenesis in large wounds	Three types of HFs: terminal, pubic/axillary, vellus Anagen duration about several years De novo HF formation up to week 23 of embryo development
Sweat glands	Only in the paw pads	Distributed along all skin regions
Wound healing and fibrosis	Skin contraction	Re-epitalization, granulation tissue formation
Immune cells	DETC are present γδT-cells are scattered throughout the dermis at a distance from aβT-cells and blood vessels	DETC are absent γδT-cells are located in clusters together with αβ-T-cells

## Data Availability

Not applicable.

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
