# Peer review of "Blank Spots in the Map of Human Skin: The Challenge for Xenotransplantation"

_ijms, 2023, doi:10.3390/ijms241612769_

Round 1

Reviewer 1 Report

This is a good review, in which the authors introduce the basic principles of allografting in the second part and mainly discuss the similarities and differences of mouse and human skin structure in the third part. But the title is "Blank spots in the map of human skin: the challenge for xenotransplantation". This part of the author did not elaborate, please add this aspect, because this is the focus of this article.

Author Response

Dear reviewer!

Thank you for your comments.

We have revised the manuscript and summarized blank spots concerning the function of epidermis, dermis, adipose tissue and immune cells in part 4 “New methods in xenografting to close the gaps in skin structure knowledge” along with further perspectives of method development (p.11-12, lines 467-534). Below you can find the fragment of our manuscript which contains these changes. Changes to the previous version of the manuscript are shown in yellow.

      1. New methods in xenografting to close the gaps in skin structure knowledge

The analogy between mouse and human skin structure indicates several significant knowledge gaps regarding our understanding of human skin cell composition and behavior.

Despite some discrepancies, the mouse and human epidermis contain a similar set of cell populations including LRCs and non-LRCs. Their distribution in the mouse epidermis depends on the distance from HFs, while in human skin it depends on rete ridges and inter-ridge domains [100,104]. Whether human HFs influence the spatial organization of LRCs in the epidermis or possess some other impact on epidermis maintenance and regeneration, as in mouse skin, remains questionable. Lineage-tracing experiments revealed that mouse HF bulge epithelial stem cells are involved in skin epithelialization after injury [162]; human HFs could contribute to epidermal regeneration similarly. Identification of epidermal stem cell populations and the detailed characterization of their molecular signatures and the localization throughout the epidermis  is extremely important for our understanding of the wound healing processes and developing the cell therapy to increase the regeneration [163,164].

The gap in our knowledge between human and mouse dermis is much more impressive. The separation of dermal fibroblasts subpopulations is possible only using surgical instruments that assume the inability to obtain the pure population. Single-cell RNA-seq, one of the most powerful approaches in human skin research, did not establish the differences in expression profiles between two major dermal fibroblast populations: papillary and reticular [115]. Thus, the investigation of age-related changes [165], disease-associated abnormalities [166,167], and the impact of fibroblast heterogeneity in the epidermal stem cell niche is sophisticated.

DWAT is known to participate in HF cycle regulation as well as in skin regeneration after injury [123]. Adipose tissue MSC-derived exosomes are known to stimulate fibroblast and keratinocyte proliferation, induce collagen remodeling, promote angiogenesis, and reduce inflammation [168]. However, in vivo studies of human material are needed to confirm their safety and efficiency as therapeutic agents.

A mouse skin investigation revealed the effects of resident immune cells on skin physiological functions. To reproduce human skin immunity in xenografts, a mouse immune system is suppressed and replaced by human cells. Studying the autoimmune disorders and immune barrier formation in co-transplantation model, human T-cells [169–171], NK-cells [172], PBMC [58], as well as immune tissues such as a spleen, a thymus, and a liver, have been transplanted along with skin. All the known or newly identified types of skin immune cells could be added to the xenograft. Also, this is the way to simulate the mouse immune system in human skin. It is interesting in the context of recently identified unique functions of mouse immune cells, in particular the involvement of γδT, aβT and DETCs in hair cycle regulation, WIHN and scarless wound healing [161,173].

Xenografts could contain all the cell types of human skin. Modern genetically modified immunodeficient mouse strains allow for the maintenance of up to 6 cm2 of human skin transplants for a year without rejection [21]. For the cell types that are depleted as a result of withdrawing from the human organism, like immune and endothelial cells, there is a possibility to isolate and inject the cell suspension directly into the graft or into the recipient animal [174], or to transplant the tissue that produces the necessary cells [36]. However, the small number of cell types of interest should be considered when planning further experimental procedures due to the inability to employ several approaches [174].

Xenografting allows the application of vital tracers and chemicals. Nucleotide analogs (BrdU, EdU, IdU) can be used for label-retaining or label-dilution experiments [175]; fatty acid analogues, for example, BODIPY, can be used to analyze the DWAT metabolism [124]. Lentiviral particles bearing fluorescent proteins proved their efficiency in xenograft epidermis transfection and could be helpful for clonal analysis and cell tracing experiments [93,176].

Skin transfection could also be suitable for genetic alterations in gene expression, which are obligatory for deciphering the signaling cascades involved in homeostasis maintenance and regeneration after injury, defining intracellular protein functions, and exploring potential targets for genetic therapy of skin disorders. For example, siRNA has already been used in human skin xenografts [57,176]. While lentiviral transfection is suitable for highly proliferative epidermal cells, adeno- and adeno-associated viruses could be employed for quiescent cells [177]. Hypothetically, the further generation of viral transfection could be based on tissue-specific viruses, like VZV or cytomegalovirus for dermal targeting, as was realized for orthoreovirus implementation for cancer xenografts [178]. There are several alternative techniques to deliver the genetic material, like packaged with Histidine Lysine Polymer [57], electrotransfer [179], microneedling with vectorless (naked) plasmid DNA [180].

Skin xenotransplantation allows studying changes at different terms after grafting. Combination with cell tracing methods will give us the possibility to describe the dynamical changes as it has already been done in mice.

Reviewer 2 Report

In this review article, the authors summarize xenotransplantation of human skin into animals. They also compare human skin to mouse skin and discuss current topics of skin biology. The article is well written with substantial numbers of references. However, there are some concerns in the manuscript. My concerns are listed below.

1: The major limitation of xenotransplantation is that immune system is dependent on the recipient animals and the immunological reaction to the graft is completely different from that in the real human body. This major limitation is not well discussed in the manuscript.

2: Several immunodeficient animals (Nude, SCID, NOD-SCID) are used in the xenotransplantation experiments. Are there any studies which compared the recipient animals? What should we take into consideration when we select recipient animals in a certain experiment?

3: In ‘Conclusions’, some new technologies and methods in xenotransplantation are described. However, these new technologies and methods should be mentioned in a separated section, not in ‘Conclusions’. ‘Conclusions’ should be shorter.

4: In page 9 (line 370), some references are shown with the authors’ name, not with the reference number. 

Author Response

Dear Reviewer!

Thank you for your review of our paper.

We have carefully reviewed the comments and have revised the manuscript accordingly. Our responses are given in a point-by-point manner below. Changes to the manuscript are shown in yellow.

1: The major limitation of xenotransplantation is that immune system is dependent on the recipient animals and the immunological reaction to the graft is completely different from that in the real human body. This major limitation is not well discussed in the manuscript.

The difference between mouse and human immune system as well as the reaction to the graft is a significant point. After revision we discussed this problem and the possibility of human immunological microenvironment reproduction in xenograft in part 4 “New methods in xenografting to close the gaps in skin structure knowledge” at page 11, lines 496-505:

A mouse skin investigation revealed the effects of resident immune cells on skin physiological functions. To reproduce human skin immunity in xenografts, a mouse immune system is suppressed and replaced by human cells. Studying the autoimmune disorders and immune barrier formation in co-transplantation model, human T-cells [169–171], NK-cells [172], PBMC [58], as well as immune tissues such as a spleen, a thymus, and a liver, have been transplanted along with skin. All the known or newly identified skin immune cells could be added to the xenograft. Also, this is the way to simulate the mouse immune system in human skin. It is interesting in the context of recently identified unique functions of mouse immune cells, in particular the involvement of γδT, aβT and DETCs in hair cycle regulation, WIHN and scarless wound healing [161,173].

2: Several immunodeficient animals (Nude, SCID, NOD-SCID) are used in the xenotransplantation experiments. Are there any studies which compared the recipient animals? What should we take into consideration when we select recipient animals in a certain experiment?

After revision we decided to describe mouse lines used in xenotransplantation studies in more detail (page 11, lines 95-120). We have also discussed the comparison between different recipient mouse strains in the context of skin xenografting (page 12, lines 121-133):

2.1 Animals 

The development of the appropriate mouse strain requires maximal immunity suppression and minimal risks of graft rejection. BALB/c or C/B-17 mice are usually used as a background. The first mouse strain that gave rise to the xenotransplantation method was BALB/c-nu/nu (Nude), which lacks T-cell immunity but retains B-cells, NK cells, and macrophages, therefore frequently resulting in graft rejection or scarring [23–25].

To efficiently suppress mouse immunity T and B cells should be removed. Now in most experiments mice with autosomal recessive mutation in Prkdcscid allele (SCID mutation) are used for human skin transplantation. SCID mice have impaired T and B lymphocyte development [26,27]. Rag1 or Rag 2 knockout (Rag-/-) which leads to T- and B- lymphocyte arrest in bone marrow may also be used [28].

Another important component of immunity - NK cells. NOD strain is characterized by deficient NK cell function and may be used as a background for SCID or Rag-/- mutations. NK cell deficiency may also be caused by bg mutation [29,30]. Mutation in the interleukin-2 gamma chain receptor (IL2rγ) leads to the disruption of signaling pathways that are involved in hematopoietic cell development and NK cell differentiation. Its application significantly reduces the graft infiltration and may be combined with SCID [31–33] or with Rag-/- mutations [34,35].

Different combinations of mentioned background strains and mutations may be used to improve mouse immune tolerance. For example, the NIH-III mouse strain has a Nude background and also possesses a xid mutation that affects the maturation of T-independent B lymphocytes and a bg mutation [29,30]. One of the popular strains is NOD-scid IL2rγnull or NSG that has a NOD background, SCID mutation that affects T and B cells and mutation in IL2rγ that affects other immunity components [31–33].

Several research groups prefer to use rat strains versus mice [36,37]. Rat skin morphology is closer to human skin as compared to mouse models, and rat lifespan is longer than that of mice, thus providing more prolonged experiments.

Experimental design requires an individual approach to selection of an appropriate mouse strain. The comparison between SCID/bg and NSG has shown that human skin morphology in SCID/bg mice is better reproduced compared with NSG. At the same time, co-transplantation with hematopoietic cells is more efficient at NSG strain [38]. Tumor xenograft growth dynamics are also different between various mouse strains. In such cases a panel of several mouse strains may be used [39].

Reconstruction of some human skin processes is equally successful by applying different mouse strains. For example, the pathogenesis of varicella-zoster virus (VZV) did not differ between xenografts in SCID and nude mice [23] or SCID and NOD-SCID [27].

There is a small number of studies comparing skin xenografting between several mouse strains perhaps due to technical difficulties of model construction and high cost. Therefore choice of specific mouse strain may be based on previous researches where similar conditions were studied.

3: In ‘Conclusions’, some new technologies and methods in xenotransplantation are described. However, these new technologies and methods should be mentioned in a separated section, not in ‘Conclusions’. ‘Conclusions’ should be shorter.

We separated the part concerning some new technologies and methods in xenotransplantation from the ‘Conclusions’ section. We have also summarized some blank spots in human skin structure in part 4 “New methods in xenografting to close the gaps in skin structure knowledge” (pages 11-12, lines 467-542):

4. New methods in xenografting to close the gaps in skin structure knowledge

The analogy between mouse and human skin structure indicates several significant knowledge gaps regarding our understanding of human skin cell composition and behavior.

Despite some discrepancies, the mouse and human epidermis contain a similar set of cell populations including LRCs and non-LRCs. Their distribution in the mouse epidermis depends on the distance from HFs, while in human skin it depends on rete ridges and inter-ridge domains [100,104]. Whether human HFs influence the spatial organization of LRCs in the epidermis or possess some other impact on epidermis maintenance and regeneration, as in mouse skin, remains questionable. Lineage-tracing experiments revealed that mouse HF bulge epithelial stem cells are involved in skin epithelialization after injury [162]; human HFs could contribute to epidermal regeneration similarly. Identification of epidermal stem cell populations and the detailed characterization of their molecular signatures and the localization throughout the epidermis  is extremely important for our understanding of the wound healing processes and developing the cell therapy to increase the regeneration [163,164].

The gap in our knowledge between human and mouse dermis is much more impressive. The separation of dermal fibroblasts from different subpopulations is possible only using surgical instruments that assume the inability to obtain the pure population. Single-cell RNA-seq, one of the most powerful approaches in human skin research, did not establish the differences in expression profiles between two major dermal fibroblast populations: papillary and reticular [115]. Thus, the investigation of age-related changes [165], disease-associated abnormalities [166,167], and the impact of fibroblast heterogeneity in the epidermal stem cell niche is sophisticated.

DWAT is known to participate in HF cycle regulation as well as in skin regeneration after injury [123]. Adipose tissue MSC-derived exosomes are known to stimulate fibroblast and keratinocyte proliferation, induce collagen remodeling, promote angiogenesis, and reduce inflammation [168]. However, in vivo studies of human material are needed to confirm their safety and efficiency as therapeutic agents.

A mouse skin investigation revealed the effects of resident immune cells on skin physiological functions. To reproduce human skin immunity in xenografts, a mouse immune system is suppressed and replaced by human cells. Studying the autoimmune disorders and immune barrier formation in co-transplantation model, human T-cells [169–171], NK-cells [172], PBMC [58], as well as immune tissues such as a spleen, a thymus, and a liver, have been transplanted along with skin. All the known or newly identified skin immune cells could be added to the xenograft. Also, this is the way to simulate the mouse immune system in human skin. It is interesting in the context of recently identified unique functions of mouse immune cells, in particular the involvement of γδT, aβT and DETCs in hair cycle regulation, WIHN and scarless wound healing [161,173].

Xenografts could contain all the cell types of human skin. Modern genetically modified immunodeficient mouse strains allow for the maintenance of up to 6 cm2 of human skin transplants for a year without rejection [21]. For the cell types that are depleted as a result of withdrawing from the human organism, like immune and endothelial cells, there is a possibility to isolate and inject the cell suspension directly into the graft or into the recipient animal [174], or to transplant the tissue that produces the necessary cells [36]. However, the small number of cell types of interest should be considered when planning further experimental procedures due to the inability to employ several approaches [174].

Xenografting allows the application of vital tracers and chemicals. Nucleotide analogs (BrdU, EdU, IdU) can be used for label-retaining or label-dilution experiments [175]; fatty acid analogues, for example, BODIPY, can be used to analyze the DWAT metabolism [124]. Lentiviral particles bearing fluorescent proteins proved their efficiency in xenograft epidermis transfection and could be helpful for clonal analysis and cell tracing experiments [93,176].

Skin transfection could also be suitable for genetic alterations in gene expression, which are obligatory for deciphering the signaling cascades involved in homeostasis maintenance and regeneration after injury, defining intracellular protein functions, and exploring potential targets for genetic therapy of skin disorders. For example, siRNA has already been used in human skin xenografts [57,176]. While lentiviral transfection is suitable for highly proliferative epidermal cells, adeno- and adeno-associated viruses could be employed for quiescent cells [177]. Hypothetically, the further generation of viral transfection could be based on tissue-specific viruses, like VZV or cytomegalovirus for dermal targeting, as was realized for orthoreovirus implementation for cancer xenografts [178]. There are several alternative techniques to deliver the genetic material, like packaged with Histidine Lysine Polymer [57], electrotransfer [179], microneedling with vectorless (naked) plasmid DNA [180].

Skin xenotransplantation allows studying changes at different terms after grafting. Combination with cell tracing methods will give us the possibility to describe the dynamical changes as it has already been done in mice.

      1. Conclusions

In this review, we highlighted several issues related to human skin-specific markers, proliferation potential, origin, functions, and interactions with other cell types and skin structures. The lack of answers to these questions arises due to the limited spectrum of available methods for human skin research versus mouse skin. Human skin xenografting compensates for these differences. Combining with the single cell approach, it could significantly promote our knowledge about human skin development, self-renewal, and regeneration.

4: In page 9 (line 370), some references are shown with the authors’ name, not with the reference number. 

We have corrected this point. You may check the changes in page 9, line 395:

DWAT secretes BMP2 [126], PDGF [127], Follistatin, DKK1 and SFRP4 [128] which are or-chestrates the hair cycling

Round 2

Reviewer 1 Report

Thanks for your point-to-point reply, it looks good for me

Reviewer 2 Report

The manuscript improved after revision, and is now acceptable for publication.